# Effects of Photoperiod Interacted with Nutrient Solution Concentration on Nutritional Quality and Antioxidant and Mineral Content in Lettuce

**Jiali Song, Hui Huang, Shiwei Song** **, Yiting Zhang, Wei Su and Houcheng Liu \***

College of Horticulture, South China Agricultural University, Guangzhou 510642, China;
song46@stu.scau.edu.cn (J.S.); dawnzsl@stu.scau.edu.cn (H.H.); swsong@scau.edu.cn (S.S.);
yitingzhang@scau.edu.cn (Y.Z.); susan_l@scau.edu.cn (W.S.)

**\*** Correspondence: liuhch@scau.edu.cn; Tel.: +86-020-85280464

**Abstract:** The interacted effects of photoperiod and nutrient solution concentrations (NSCs) on nutritional quality and antioxidant and mineral content in lettuce were investigated in this study. There were a total of nine treatments by three photoperiods (12 h/12 h, 15 h/9 h, and 18 h/6 h), with a combination of three NSCs (1/4, 1/2, and 3/4 NSC). The contents of photosynthetic pigment, mineral element, and nutritional quality were markedly affected by the combination of photoperiod and NSC. The highest leaf number and plant weight were found in lettuce under the combination of 18–0.25X. There was a higher content of photosynthetic pigment in treatment of 15-0.25X. Shorter photoperiod (12 h/12 h and 15 h/9 h) and NSC (1/4 and 1/2 NSC) contributed to reduced nitrate contents and higher contents of free amino acid, soluble protein, and vitamin C. Longer photoperiod and lower NSC could increase soluble sugar content. The content of total P, K, and Ca exhibited a similar trend under the combination of photoperiod and NSC, with a higher content at 3/4 NSC under different photoperiods. Lower contents of total Zn and N were found under longer photoperiod. Moreover, higher antioxidant contents, including 2, 2-diphenyl-1-picrylhydrazyl (DPPH), value of ferric-reducing antioxidant power (FRAP), flavonoid, polyphenol, and anthocyanin were observed under shorter photoperiod, with the peak under 12-0.50X. Generally, 12-0.50X might be the optimal treatment for the improvement of the nutritional quality of lettuce in a plant factory that produced high-quality vegetables.

**Keywords:** photoperiod; nutrient solution; antioxidants; phytochemical; lettuce; biomass

## 1. Introduction

The nutritional qualities of vegetables possess potential benefits for human health, including soluble solid, antioxidant activity, and carotenoid. These secondary metabolites not only suppress a variety of diseases but also enhance antioxidation ability [1,2]. In plant factories, temperature, light and nutrient solution are the main influencers of the nutritional quality of vegetables. Artificial lighting is the most important factor during plant growth. Photoperiod could affect growth and metabolism in many vegetables, such as accumulation of nutritional compound, biomass, and pigment [3,4]. Photoperiod was highly significant, positively related to biomass at the same total daily light input [5]. Moreover, longer photoperiod increases the content of crude fiber, vitamin C, soluble sugar, free-radical scavenging activity, phenolic compound content, and anthocyanin and decreases nitrate content [5–7], whereas 12 h/12 h photoperiod could increase the contents of antioxidant activity, total polyphenol, and chlorophyll in leafy vegetables [3]. Fresh and dry mass production in kale (*Brassica oleracea* L. var. *acephala* D.C) increased linearly with increasing photoperiod and reached the maximum under

the 24 h photoperiod [4]. However, a long photoperiod could improve dry weight, fresh weight, leaf chlorophyll content, and the leaf area in radish until it extends to 24 h, when these effects are reduced [8–10].

Hydroponic systems are the primary systems in plant factories. Therefore, the reasonable arrangements of nutrient solution are one way to modify the quality and yield of vegetables as well as reducing cost. A previous study has demonstrated that a lower concentration of nutrient solution could result in the reduction of biomass in green and red Salanova® butterhead lettuce, but a moderate concentration of nutrient stress (0.75 dS m$^{-1}$) could improve the content of anthocyanins, total phenolic acids, caffeoyl-meso-tartaric, chicoric, chlorogenic, and total ascorbic acid in red Salanova® butterhead lettuce [11]. The content of total soluble solid and sugar in cherry tomato at continuous high electrical conductivity (EC) (4.7 dS·m$^{-1}$) is higher than continuous low EC (2.8 dS·m$^{-1}$) [12,13]. The growth and quality of hydroponic vegetables are affected by integrating factors in closed plant factory, such as light, nutrient supply, and temperature. High light illumination combined with low nitrogen is beneficial for reducing nitrate content and increasing vitamin C content [14]. Longer photoperiod and higher temperature are conducive to the formation of garlic bulb and improvement of quality [15].

Did the combination of photoperiod and nutrient solution concentrations (NSC) regulate lettuce growth and phytonutrients in lettuce? This study provides further understanding on the optimal combination of photoperiod and NSC for the improvement of quality and yield of lettuce through investigating the effects of antioxidant activity, mineral element content, and nutritional quality in lettuce under different photoperiod and nutrient solution conditions.

## 2. Materials and Methods

### 2.1. Plant Materials

This study was carried out in South China Agricultural University with lettuce (*Lactuca sativa* L. cv. Italy). When the lettuce seedlings grown at 1/4 NSC in sponge block showed three expended true leaves, they were removed in a growth chamber with hydroponic system under a relative humidity of 60–80%, day/night of 22–25/14–18 °C, ambient $CO_2$ concentration, the nutrient solution refreshed every 10 days, and aeration rate of 15min/h. The elements in full-strength Hoagland solutions are listed in Table S1.

### 2.2. Treatments

Treatments of three photoperiods (12 h/12 h, 15 h/9 h, and 18 h/6 h) combined with three NSCs (1/4, 1/2, and 3/4) were performed (Table 1 and Table S1). The photosynthetic photon flux density (PPFD) was 250 μmol·m$^{-2}$·s$^{-1}$ based on the mixed light-emitting diode (LED, red: blue = 2:1; blue: 460 ± 10 nm, red: 660 ± 10 nm) in all treatments (Figure S1). The lettuces at 30 days after transplant were sampled and the statistical data of leaf number per plant were collected. The lettuce samples were oven-dried at 75 °C for 48 h. Lettuce dry and fresh weight were measured by electronic balance.

**Table 1.** Nine treatments of photoperiod combined with nutrient solution concentration.

| Photoperiod \ NSC | 1/4 | 1/2 | 3/4 |
|---|---|---|---|
| 12 h/12 h | 12-0.25X | 12-0.50X | 12-0.75X |
| 15 h/9 h | 15-0.25X | 15-0.50X | 15-0.75X |
| 18 h/6 h | 18-0.25X | 18-0.50X | 18-0.75X |

### 2.3. Phytochemical Measurements

Chlorophyll content was determined colorimetrically by anthrone test [16]. Carotenoid and chlorophyll a and b were extracted by 25 mL acetone alcohol mixture using 0.5 g fresh levels and

measured at 440 nm, 645 nm, and 663 nm by UV spectrophotometer (Shimadzu UV-16A, Shimadzu, Japan). The chlorophyll content was calculated according to Song et al. [17].

Soluble protein content was measured according to Bradford et al. [18]. 0.5 g fresh lettuce was ground up by a mortar and pestle with liquid nitrogen, and then added into 5 mL distilled water. A total of 0.1 mL supernatant, which was acquired from the centrifuged solution (10000 rpm for 10 min at 4 °C), was added into 4.9 mL Coomassie brilliant blue G-250 solution (Sigma, Olympia, WA, USA, 0.1 g·L$^{-1}$), after which was examined by UV-spectrophotometer at 595 nm.

Nitrate content was examined according to the method of Cataldo et al. [19]. A total of 1 g fresh lettuce was boiled for 30 min with 10 mL distilled water. Then, 9.5 mL NaOH (8%) and 0.4 mL salicylic and sulfuric acid (5%) were added into 0.1 mL extracting solution. The absorbance of mixture was detected at 410 nm by UV spectrophotometer.

The contents of vitamin C were measured according to the method of Shyamala et al. [20]. A total of 0.5 g fresh lettuce was ground up by a mortar and pestle with the mixed solution (1 mL 15% potassium ferrocyanide, 1mL zinc sulfate (30%) and 3 mL oxalic acid (1%), 4 mL ammonium molybdate, 2 mL vitriol (5%), and 1 mL phosphate-acetic acid were added. The content of vitamin C was measured by UV spectrophotometer at 500 nm.

The content of soluble sugar was determined according Song et al. [21]. A total of 0.5 g fresh lettuces were boiled for 30 min with 10 mL distilled water. Then 5 mL vitriol, 0.5 mL anthrone ethyl acetate, and 1.9 mL distilled water were added into 0.1 mL supernatant. The content of soluble sugar was measured at 630 nm by UV spectrophotometer.

The contents of free amino acid were examined according the method of Cataldo et al. [19]. A total of 1 g fresh lettuce was heated in a water bath for 30 min at 80 °C with deionized water (10 mL) and centrifuged for 10 min at 13,000 g. Then 19 mL NaOH (4 mol·L$^{-1}$) and 0.8 mL salicylic acid (5% (*w/v*), Sigma, Olympia, WA, USA) was added into 0.2 mL supernatant. The absorbance of mixture was examined at 410 nm by UV spectrophotometer.

Polyphenol contents were examined according to Tadolini et al. [22]. A total of 1 g fresh lettuce was ground up by a mortar and pestle liquid nitrogen, and then added into 8 mL methanol (80%). After 60 min in a boiling bath, the supernatants that had been centrifuged for 10 min at 12,000 rpm were transferred into evaporation flask at 40 °C. The exacting solution with distilled water (10 mL) was centrifuged for 20 min at 8000 rpm. A total of 11.5 mL sodium carbonate (26.7%), 0.5 mL foline-phenol, and 7ml distilled water were added to 1 mL supernatant. The absorbance of polyphenol was determined by UV spectrophotometer at 760 nm.

Anthocyanin contents were determined according to Rapisarda et al. [23]. A total of 1.0 g fresh lettuces were boiled for 2 h with 20 mL 60% alcohol. The absorbance of anthocyanin was examined at 535 nm by UV spectrophotometer.

Flavonoid contents were examined using the spectrophotometric method [24]. The mixed solution of 11.5 mL alcohol (30%) and 0.7 mL NaNO$_2$ (5%) was added to 1 mL extract solution. Five minutes later, 0.7 mL Al (NO$_3$)$_3$ (10%) was added, and then after 6 min, 5 mL NaOH (5%) was added. The absorbance of mixture was performed by UV spectrophotometer at 510 nm.

The DPPH radical scavenging rate was examined according to the method of Tadolini et al. [22]. A 0.5 g sample was added into 2.5 mL DPPH solution (65 μmol·L$^{-1}$). Approximately 30 min later, the absorbance was varied out at 517 nm by UV spectrophotometer.

The FRAP was examined basing on the method of Benzie and Strain [25]. A total of 3.6 mL of the mixed solution containing acetate buffer (0.3 mol·L$^{-1}$), TPTZ (10 mmol·L$^{-1}$), and FeCl$_3$ (20 mmol·L$^{-1}$) = 10:1:1 was added into the 0.4 mL sample solution. The mixture was left at 37 °C for 10 min. The absorbance was measured by UV spectrophotometer at 593 nm.

## 2.4. Mineral Element Determination

Dry samples of lettuce were used for mineral element determination. The contents of total K, P, and N were measured according to the flame photometry method [26], Mo–Sb colorimetry [27],

and Ojeda's [28]. The contents of total Ca, Mg, and Zn were examined according to the method of atomic absorption spectrophotometry [29].

*2.5. Data Analysis*

One-way analysis of variance (one-way ANOVA) and two-way ANOVA analysis were carried out to assay the significant differences in single photoperiod/NSC factor and the interaction using SPSS17.0 software at $p \leq 0.01$ level and $p \leq 0.05$. All the assays were carried out in triplicates.

## 3. Results

*3.1. Growth and Biomass*

Longer photoperiod was favorable for lettuce biomass at lower NSC (1/4 and 1/2 NSC) (Table 2 and Figure S2). The leaf number and biomass of lettuce at 3/4 NSC treatment increased at first, and then decreased with the extension of photoperiod. The maximum leaf number was exhibited at 1/4 NSC under 12 h/12 h and 18 h/6 h photoperiod. Lettuce dry and fresh weight under the treatment of 18-0.25X were the highest. Photoperiod exhibited a prominent difference for dry weight (DW) and fresh weight (FW) of plant and shoot, whereas NSC had less effect on these growth parameters. Thus, photoperiod had more influence on lettuce growth than NSC, and the treatment of 18-0.25X contributed to lettuce production in a plant factory.

**Table 2.** Changes of growth lettuce under the interaction of photoperiod and nutrient solution concentrations (NSC).

| Treatments | | Leaf Number | Weight (g per Plant) | | | |
|---|---|---|---|---|---|---|
| Photoperiod | NSC | | Plant FW | Shoot FW | Plant DW | Shoot DW |
| 12 h/12 h | 1/4 | 19.3 ± 0.9a | 62.47 ± 2.40e | 56.07 ± 1.91f | 3.09 ± 0.18d | 2.58 ± 0.16d |
| | 1/2 | 18.3 ± 0.3ab | 71.97 ± 1.22cde | 64.53 ± 0.86def | 3.34 ± 0.03cd | 2.81 ± 0.05cd |
| | 3/4 | 18.3 ± 0.3ab | 78.30 ± 1.35bc | 72.37 ± 1.76bcd | 3.52 ± 0.06cd | 3.07 ± 0.08c |
| 15 h/9 h | 1/4 | 17.7 ± 0.7ab | 65.43 ± 1.12de | 59.77 ± 0.74ef | 3.31 ± 0.06cd | 2.81 ± 0.02cd |
| | 1/2 | 18.7 ± 0.7a | 88.10 ± 1.10b | 78.97 ± 0.95b | 4.09 ± 0.05ab | 3.48 ± 0.06b |
| | 3/4 | 18.7 ± 0.9a | 85.87 ± 2.90b | 80.53 ± 3.74b | 4.13 ± 0.10ab | 3.59 ± 0.10ab |
| 18 h/6 h | 1/4 | 19.0 ± 0.0a | 100.67 ± 6.93a | 92.40 ± 6.40a | 4.49 ± 0.26a | 3.93 ± 0.26a |
| | 1/2 | 18.3 ± 0.9ab | 86.83 ± 3.88b | 77.73 ± 3.86bc | 4.13 ± 0.22ab | 3.51 ± 0.13b |
| | 3/4 | 16.3 ± 0.3b | 74.70 ± 4.16cd | 67.90 ± 3.92cde | 3.67 ± 0.15bc | 3.03 ± 0.14c |
| ANOVA (*F* value) | Photoperiod | NS | ** | ** | ** | ** |
| | NSC | NS | NS | NS | NS | NS |
| | Photoperiod interacted with NSC | NS | ** | ** | * | ** |

Data are mean ± standard error. * and ** indicate significant difference at $p \leq 0.05$ and $p \leq 0.01$, respectively. The letters marked in tables mean the significance of difference. ($p \leq 0.05$, Tukey's test). FW = fresh weight, DW = dry weight, and NS = no significance.

*3.2. Photosynthetic Pigment Content*

The highest content of chlorophyll a, b, and carotenoid was found under 15-0.25X, and the lowest content of chlorophyll a, b, and carotenoid was exhibited under 15-0.50X. However, the highest value of chlorophyll a/b was observed under 15-0.50X (Table 3). Photoperiod, NSC, and their combinations had significant effects on contents of chlorophyll a, b, and carotenoid and the value of chlorophyll a/b (with the exception of chlorophyll a/b under the photoperiod) (Table 3). Hence, the combination of 15-0.25X contributed to accumulate the contents of carotenoid, total chlorophyll, and chlorophyll a and b, while the 15-0.50X treatment contributed to enhanced chlorophyll a/b.

**Table 3.** The effect of photosynthetic pigment under the interaction of photoperiod and NSC.

| Treatments | | Photosynthetic Pigment Content (mg/g) | | | | Chlorophyll a/b |
|---|---|---|---|---|---|---|
| Photoperiod | NSC | Chlorophyll a | Chlorophyll b | Total Chlorophyll | Carotenoid | |
| 12 h/12 h | 1/4 | 0.57 ± 0.02ab | 0.17 ± 0.01b | 0.14 ± 0.01ab | 0.74 ± 0.02a | 3.37 ± 0.06de |
| | 1/2 | 0.41 ± 0.01d | 0.12 ± 0.00e | 0.10 ± 0.00d | 0.53 ± 0.01d | 3.53 ± 0.03bc |
| | 3/4 | 0.53 ± 0.01b | 0.16 ± 0.00c | 0.13 ± 0.00bc | 0.70 ± 0.01b | 3.45 ± 0.05cde |
| 15 h/9 h | 1/4 | 0.60 ± 0.01a | 0.18 ± 0.00a | 0.14 ± 0.00a | 0.79 ± 0.01a | 3.33 ± 0.03e |
| | 1/2 | 0.32 ± 0.00g | 0.09 ± 0.00g | 0.08 ± 0.00f | 0.41 ± 0.01f | 3.69 ± 0.01a |
| | 3/4 | 0.37 ± 0.01e | 0.11 ± 0.00ef | 0.09 ± 0.00e | 0.48 ± 0.01e | 3.43 ± 0.03cde |
| 18 h/6 h | 1/4 | 0.35 ± 0.02ef | 0.10 ± 0.00f | 0.09 ± 0.00ef | 0.46 ± 0.02e | 3.41 ± 0.08cde |
| | 1/2 | 0.47 ± 0.02c | 0.13 ± 0.00d | 0.12 ± 0.01c | 0.61 ± 0.02c | 3.52 ± 0.06bcd |
| | 3/4 | 0.32 ± 0.01fg | 0.09 ± 0.00g | 0.08 ± 0.00ef | 0.42 ± 0.01f | 3.63 ± 0.04ab |
| ANOVA | Photoperiod | ** | ** | ** | ** | NS |
| (*F* value) | NSC | ** | ** | ** | ** | ** |
| | Photoperiod interacted withNSC | ** | ** | ** | ** | ** |

Data are mean ± standard error. ** indicate significant difference at $p \leq 0.01$,. The letters marked in tables mean the significance of difference. ($p \leq 0.05$, Tukey's test) and NS = no significance.

### 3.3. The Content of Free Amino Acid, Nitrate, Vitamin C, and Soluble Sugar and Protein

Two-way ANOVA confirmed that the contents of soluble protein ($p \leq 0.01$) and sugar ($p \leq 0.05$), vitamin C ($p \leq 0.01$), nitrate ($p \leq 0.01$), and free amino acid ($p \leq 0.01$) were markedly affected by NSC and photoperiod, with the exception of nitrate content under photoperiod. The combination of photoperiod and NSC was significantly responsive to the accumulation of soluble protein ($p \leq 0.01$), free amino acid ($p \leq 0.01$), and nitrate ($p \leq 0.05$) (Table S2). Higher contents of soluble sugar in lettuce were observed at 1/4 NSC, with the maximum content under 18-0.25X. The accumulation of soluble sugar generally possessed a decreasing trend with increasing NSC but an increasing trend with the extension of photoperiod (Figure 1). Nitrate content was positively affected by increasing NSC under shorter photoperiod (12 h/12 hand 15 h/9 h) (Figure 1). The highest nitrate content was found under 15-0.75X, and the lowest nitrate contents were observed under 12-0.25X. A declining trend of vitamin C content was showed with increasing NSC (Figure 1). The content of free amino acid increased and then decreased with increasing NSC level, peaking under 12-0.50X and 15-0.50X (Figure 1). The greatest content of soluble protein was exhibited under 12-0.25X and 12-0.75X, 15-0.25X and 18-0.50X, and 18-0.75X (Figure 1).

In general, the accumulation of soluble sugar benefited from 18-0.25X and the second from 12-0.25X and 15-0.25X. 12-0.25X, 15-0.25X, and 18-0.25X contributed to accumulated contents of soluble protein, vitamin C, and reduced nitrate contents in lettuce; 12-0.50X and 15-0.50X contributed to accumulation of free amino acid. It was feasible that shorter photoperiod (12 h/12 h and 15 h/9 h) combined with lower NSC (1/4 or 1/2 NSC) could reduce nitrate contents and increase the contents of free amino acid, vitamin C, and soluble protein and sugar in lettuce.

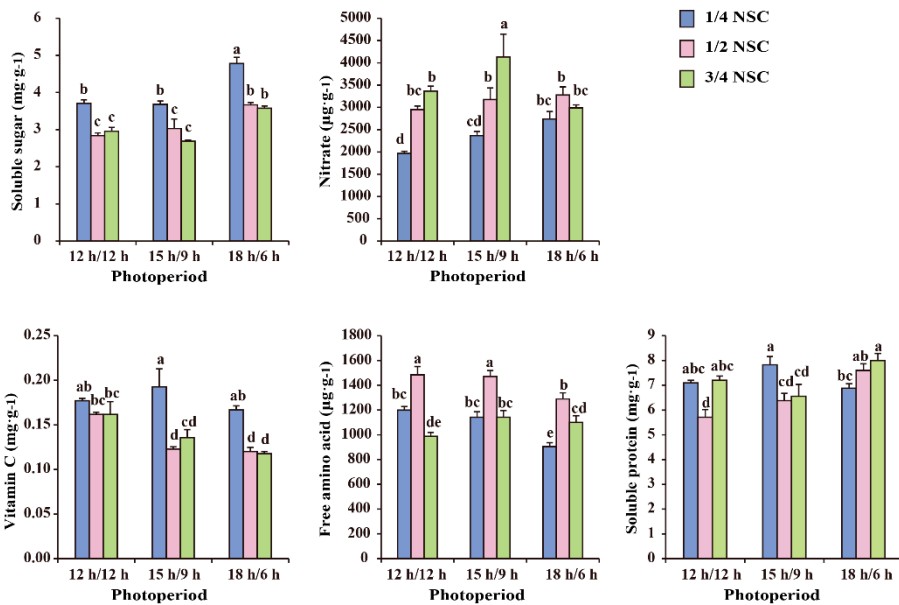

**Figure 1.** Soluble sugar, nitrate, vitamin C, soluble protein, and free amino acid contents in lettuce under the interaction of photoperiod and NSC. The letters marked in all figures mean the significance of differences ($p \leq 0.05$, Tukey's test).

### 3.4. The Content of Mineral Element

The total N content increased and then decreased with increasing photoperiod, with the highest content under 15-0.50X (Figure 2). The contents of total Ca, K, and P exhibited a similar accumulation trend under the combination of photoperiod and NSC. These increased with increasing NSC and increased or slightly decreased with the extension of photoperiod (Figure 2). The highest contents of total P were found in the treatment of 12-0.50X and 12-0.75X (Figure 2). The greatest content of total K and Ca was showed under 18-0.75X, while the lowest content was observed under 18-0.25X (Figure 2). The total Mg content was affected by photoperiod and reduced with increasing NSC, with the highest contents found under 15-0.25X (Figure 2). The lowest contents of total Zn were found under 18-0.25X, and the highest content was showed under 12-0.75X (Figure 2). Two-way ANOVA demonstrated that photoperiod, NSC, and their combinations exhibited significant effects on the contents of total P, Mg, Ca, K, Zn, and N in lettuce (Table S3) ($p \leq 0.01$).

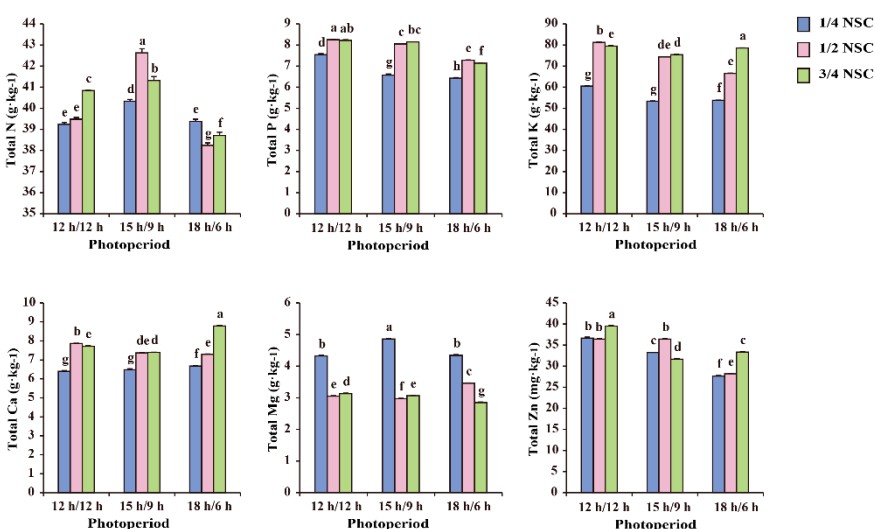

**Figure 2.** Total N, P, K, Ca, Mg, and Zn contents in lettuce under the interaction of photoperiod and NSC. The letters marked in all figures mean the significance of differences ($p \leq 0.05$, Tukey's test).

### 3.5. Antioxidant Component Content and Capacity

Two-way analysis revealed that the value of DPPH and FRAP and the contents of flavonoid polyphenol and anthocyanin in lettuce were dramatically associated with photoperiod ($p \leq 0.01$), NSC ($p \leq 0.01$), and their combinations ($p \leq 0.01$), except that flavonoid and FRAP were significantly affected by NSC ($p \leq 0.05$), and polyphenol had no significant difference at different NSC (Table S4). The content of flavonoid, polyphenol, and anthocyanin and the value of DPPH increased and then decreased with increasing photoperiod at 1/4 NSC, whereas these reduced with extending photoperiod at 1/2 and 3/4 NSC (Figure 3). The antioxidant component contents were less affected by NSC under the longest photoperiod. Therefore, the maximal contents of antioxidant component were showed in the treatment of 12-0.50X (Figure 3). These results indicated that 12-0.50X was the optimal combination for enhancement of antioxidant capacity and content in lettuce.

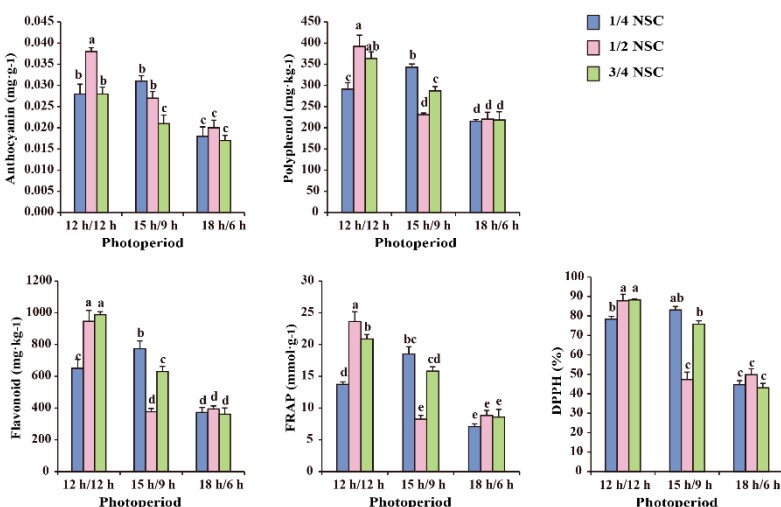

**Figure 3.** Anthocyanin, polyphenol, and flavonoid content and FRAP and DPPH in lettuce under the interaction of photoperiod and NSC. The letters marked in all figures mean the significance of differences ($p \leq 0.05$, Tukey's test).

The correlation between FRAP, DPPH, and antioxidant content in lettuce under the combination of photoperiod*NSC was analyzed (Table 4). The contents of polyphenol ($p \leq 0.01$), flavonoid ($p \leq 0.01$), DPPH ($p \leq 0.01$), and anthocyanin ($p \leq 0.01$) were greatly relevant to FRAP, and the highest coefficient was observed in flavonoid content ($r = 0.965$). The coefficient of DPPH associated with contents of polyphenol ($r = 0.918$, $p \leq 0.01$) and flavonoid ($r = 0.936$, $p \leq 0.01$) was higher than anthocyanin content ($r = 0.690$, $p \leq 0.01$). Therefore, the antioxidant activity in lettuce was primarily derived from anthocyanin, polyphenol, and flavonoid under photoperiod*NSC.

**Table 4.** The correlation analysis of antioxidant component and capacity affected by the interaction of photoperiod and NSC.

| Parameter | Polyphenol | Flavonoid | Anthocyanin | DPPH | FRAP |
|---|---|---|---|---|---|
| Polyphenol | 1 | | | | |
| Flavonoid | 0.942 ** | | | | |
| Anthocyanin | 0.748 ** | 0.722 ** | | | |
| DPPH | 0.918 ** | 0.936 ** | 0.690 ** | | |
| FRAP | 0.963 ** | 0.965 ** | 0.727 ** | 0.942 ** | 1 |

Significant difference was performed by SPSS 17.0. ** indicate significant difference at $p \leq 0.01$.

## 4. Discussion

Previous studies have demonstrated that the extension of photoperiod could increase leaf area, leaf chlorophyll content, fresh weight, and dry weight in radish, but these effects reduced when photoperiod extended to 24 h [8–10]. The lettuce biomass under 24 h/400 μmol·m$^{-2}$·s$^{-1}$ (illumination time/light intensity) was dramatically higher than 16 h/600 μmol·m$^{-2}$·s$^{-1}$ when the total daily light input of the two treatments was kept the same [5]. In this study, longer photoperiod contributed to the growth and biomass of lettuce at lower NSC, showing the highest in the treatment of 18-0.25X (Table 2). A long photoperiod produced more photosynthate, which might be due to more light energy input. Lower nutrient solution could reduce lettuce biomass [11]. However, the leaf number and fresh and dry weight of lettuce were less significantly affected by different NSC under 12 h/12 h and 15 h/9 h photoperiods (Table 2). Thus, the combination of 18-0.25X was likely the suitable condition for plant growth and biomass of lettuce. Pigment accumulations were significantly affected by different photoperiod in kale, with the maximum content of chlorophyll a and b under the 12 and 24 h photoperiod, respectively [4]. In this study, the highest pigment content was observed under 15-0.25X (Table 3). Higher chlorophyll contents were also affected by higher fertility treatment in Chinese kale [30]. However, the maximum value of chlorophyll a/b and the highest contents of chlorophyll a, b, and carotenoid in lettuce were found under 15 h/9 h photoperiod at lower NSC treatments (1/4 and 1/2 NSC). It was likely that the interaction of photoperiod and NSC had effects on chlorophyll contents (Table 3).

Longer photoperiod could achieve higher content of vitamin C and soluble sugar and protein in lettuce [5,6]. Similarly, the accumulation of soluble sugar content exhibited an increasing trend with the extension of photoperiod, while a decreasing trend with increasing NSC was observed (Figure 1). The content of free amino acid, vitamin C, and soluble protein was relatively higher under lower photoperiod (12 h/12 h and 15 h/9 h) and NSC (1/4 and 1/2 NSC) (Figure 1). Nitrate content accumulated in lettuce leaves was mainly affected by lighting and fertilizer usage during cultivation [31,32]. Longer photoperiod could reduce nitrate content in lettuce [5,6]. However, there were no significant differences in nitrate content under different photoperiod (Table S2). That was probably because the accumulation of nitrate was also associated with light intensity and light quality [5,6,17]. Increasing N in nutrient solutions could lead to the improvement of nitrate content in lettuce [14], while it showed a lower level at medium EC level in pakchoi [33]. In this study, nitrate content also exhibited an increasing tendency with increasing NSC (Figure 1). Therefore, it was likely that the combination of lower photoperiod (12 h/12 or 15 h/9 h) and 1/4 NSC or 1/2 NSC contributed to the improvement of nutrition quality in lettuce.

Mineral nutrient was important for carbohydrate content and photosynthesis in plant [34]. The mineral contents showed significant difference in vegetables under different light condition. Higher content of total N, Zn K, Mg, Ca, and P was observed under low light intensity [35–37], while they were also dramatically affected by different light quality in vegetables [38,39]. Lettuce treated by 24 h/0 h photoperiod in 30 days could significantly reduce the contents of total Zn, Cu, Fe, Mn, Mg, and Ca [40]. In this study, the contents of total Ca, K, and P increased or slightly decreased with the extension of photoperiod (Figure 2), and higher contents of total Zn and N were found under shorter photoperiod (Figure 2). The total Mg content was negatively accumulated with increasing NSC, reaching the maximum under 15-0.25X (Figure 2). Additionally, the content of mineral element was significantly responsive to different NSC (Table S3), which was consistent with our previous study [17].

Some phytochemicals were of vital importance for human health, such as anthocyanin, polyphenol, and flavonoid [41–44]. In plant factories, phytochemicals optimization could be achieved via the reasonable design of light conditions [45]. Previous studies have demonstrated that 12 h/12 h photoperiod contributed to increased contents of antioxidant activity, total polyphenol, chlorophyll, and betacyanin in five leafy vegetables (green amaranth, red beet, swiss chard, red spinach, and red amaranth) [3], whereas longer photoperiod could obtain higher anthocyanin content [6]. Continuous 24 h LED light also dramatically enhanced free-radical scavenging activity and phenolic compound

content in lettuce [7]. In this study, higher contents of DPPH, FRAP, flavonoid, anthocyanin, and polyphenol were observed under shorter photoperiod (Figure 3). The contents of flavonoid and polyphenol increased at lower fertilizer concentrations in plant [46]. The content of anthocyanin and polyphenol was improved in radish under lower solution concentration [47]. In this study, the antioxidant capacity reached the peak under the combination of 12-0.50X (Figure 3). However, it was higher at the lowest nutrient solution level (1/4 NSC) under different light intensity in our previous study [17]. It was probably because the antioxidant capacity in lettuce was significantly associated with the combination of photoperiod and NSC ($p \leq 0.01$) (Table S4). Therefore, the treatment of 12-0.50X contributed to greatly improving the antioxidant content and capacity in lettuce. However, further research is still needed to understand how the interaction of photoperiod and nutrient concentration affects plant growth and development.

## 5. Conclusions

Different nutrient solutions and light conditions are closely associated with photosynthesis, morphology, and plant growth in plant factories. 18-0.25X showed remarkable effects on improving growth of lettuce; the combination of lower photoperiod (12 h/12 or 15 h/9 h) and NSC (1/4 NSC or 1/2 NSC) was conducive to reduced nitrate content and increased content of free amino acid, vitamin C, and soluble protein and sugar;12-0.50X improved the antioxidant content and capacity. Therefore, the 18-0.25X treatment was favorable for lettuce production, while the treatment of 12-0.50X was the optimal condition for nutritional quality of lettuce. Overall, it is likely that 12-0.50X was more suitable for plant factories that produces high-quality vegetables.

**Supplementary Materials:** The following are available online at http://www.mdpi.com/2073-4395/10/7/920/s1, Figure S1: The Spectral distributions of LEDs in this study, Table S1: The concentration of mineral element in different NSC; Table S2: Two-way ANOVA analysis of nutritional quality accumulated under the interaction of photoperiod and NSC; Table S3: Two-way ANOVA analysis of mineral element accumulated under the interaction of photoperiod and NSC; Table S4: Two-way ANOVA analysis of antioxidant component accumulated under the interaction of photoperiod and NSC.

**Author Contributions:** J.S. and H.H. performed the experiment and participated in the data analysis. W.S., Y.Z. and S.S. drafted the manuscript. H.L. conceived of the study and participated in its design. H.L. acquired funding and helped to draft the manuscript. All authors read and approved the final manuscript.

**Funding:** This work was supported by the National Key Research and Development Program of China (2017YFD0701500), Key Research and Development Program of Guangdong (2019B020214005), and the Guangzhou Science & Technology Project (201704020058).

**Conflicts of Interest:** The authors declare no conflicts of interest.

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
