# Peer review of "Effects of Photoperiod Interacted with Nutrient Solution Concentration on Nutritional Quality and Antioxidant and Mineral Content in Lettuce"

_agronomy, doi:10.3390/agronomy10070920_

Round 1

Reviewer 1 Report

Article

Effects of photoperiod interacted with nutrient solution concentration on nutritional quality, antioxidant and mineral content in lettuce

Jiali Song, Hui Huang, Shiwei Song, Yiting Zhang, Wei Su and Houcheng Liu*

lighting is the  most important factor during plant growth. Photoperiod could affect growth and metabolism in  many vegetables, such as accumulation of nutritional compound, biomass and pigment.  Photoperiod was highly significant positively related to biomass at the same total daily light input. Moreover, longer photoperiod increases the content of crude fiber, vitamin C and soluble sugar, and decrease nitrate content.

In this study, higher contents of DPPH, FRAP, flavonoid, anthocyanin and polyphenol were observed under shorter photoperiod. The contents of flavonoid and polyphenol increased at lower fertilizer concentrations in plant . The content of anthocyanin and polyphenol was improved in radish under lower solution concentration. In this study, the antioxidant capacity reached the peak under the combination of 12 h/12 h*1/2NSC. However, it was higher at the lowest nutrient solution level (1/4NSC) under different light intensity in our previous study.

The content of anthocyanin and polyphenol was improved in radish under lower solution concentration. In yours study, the antioxidant capacity reached the peak under the combination of 12 h/12 h*1/2NSC. However, it was higher at the lowest nutrient solution level (1/4NSC) under different light intensity in yours previous study.

The results are very useful and continue your research

Author Response

Effects of photoperiod interacted with nutrient solution concentration on nutritional quality, antioxidant and mineral content in lettuce

Jiali Song, Hui Huang, Shiwei Song, Yiting Zhang, Wei Su and Houcheng Liu*

lighting is the most important factor during plant growth. Photoperiod could affect growth and metabolism in many vegetables, such as accumulation of nutritional compound, biomass and pigment.  Photoperiod was highly significant positively related to biomass at the same total daily light input. Moreover, longer photoperiod increases the content of crude fiber, vitamin C and soluble sugar, and decrease nitrate content.In this study, higher contents of DPPH, FRAP, flavonoid, anthocyanin and polyphenol were observed under shorter photoperiod. The contents of flavonoid and polyphenol increased at lower fertilizer concentrations in plant. The content of anthocyanin and polyphenol was improved in radish under lower solution concentration. In this study, the antioxidant capacity reached the peak under the combination of 12 h/12 h*1/2NSC. However, it was higher at the lowest nutrient solution level (1/4NSC) under different light intensity in our previous study. The content of anthocyanin and polyphenol was improved in radish under lower solution concentration. In yours study, the antioxidant capacity reached the peak under the combination of 12 h/12 h*1/2NSC. However, it was higher at the lowest nutrient solution level (1/4NSC) under different light intensity in yours previous study. The results are very useful and continue your research.

Answers: Thank you for pointing out this. Antioxidant capacity is also affected by light intensity, light quality and photoperiod. The antioxidant capacity reached the peak under the combination of 350μmol·m-2·s-1*1/4NSC in our previous study but under the light intensity of 250μmol·m-2·s-1 in this study.

Reviewer 2 Report

Effects of photoperiod interacted with nutrient solution concentration in lettuce has been studied. For the long-day plant, the longer the photoperiod the higher the biomass (Within the limit of photosynthesis). Compared with photoperiod, light intensity and quality have more influence here. DLI seems to make more sense than a simple photoperiod. It is suggested that the author take full consideration in designing the experiment in the future. The following contents are recommended for major modification.

Abstract:

Line 23-25:

So, in the practical production, you choose biomass or nutritional quality? That doesn't seem to be the final conclusion of this study.

Introduction:

The effects of photoperiod and nutrient concentration on plants and their importance should be introduced respectively, and more importantly is that why study the interaction between these two factors should be described in detail.

Some introductions should be added for the amino acid, polyphenol, anthocyanin et al that several compounds analyzed in this expt.

Materials and methods 

2.1 should be plant material and cultivation environment, and this section is not adequately described.

Line 62:

Growth chamber or plant factory? Described “plant factory” in abstract.

Line 63:

“the nutrient solution refreshed every 10 days”: The EC changes every day as plants grow. For the experiment which studied the concentration of nutrient solution, should the same EC and pH be maintained every day?

Line 64:

Even though it has table s1, but better indicate the ratio of NPK here.

Line 66-67:

NSC is nutrient solution concentrations, should remove the concentrations in “combined with three concentrations of NSC”. Remove NSC in “(1/4, 1/2 and 3/4 NSC)”.

Table 1:

The design of this table is meaningless. The following table and figures also do not continue this representation.

I recommend use the number of treatments: T1,T2…… or shorten the treatment name: 12-0.25x, 15-0.5x……..

2.5:

No information on replication was found for this expt.

Results:

3.1:

There is no description of this section in the M&M.

Table 2&3:

Are there no significant differences in other basic parameters of growth and development? For example: shoot & root length, leaf area, root biomass………..

It doesn't make sense to leave two decimal places on the number of leaves.

The decimal point & letters mark should be aligned & recommend to remove the standard error to show the data more clearly.

For “ANOVA (F value)” recommend use the common notation: NS,*,**,***.

Figure 1,2&3:

The vertical coordinate already clearly represents the content of the data, so A, B, C… can be removed.

Use standard representation for all units: mg/g → mg·g-1

Discussion & Conclusions

The interaction between photoperiod and nutrient concentration should be further discussed.

The importance of biomass and nutrient content can be discussed according to market demand, and then recommend one of the best results to the reader or grower.

Author Response

Comments and Suggestions for Authors

Effects of photoperiod interacted with nutrient solution concentration in lettuce has been studied. For the long-day plant, the longer the photoperiod the higher the biomass (Within the limit of photosynthesis). Compared with photoperiod, light intensity and quality have more influence here. DLI seems to make more sense than a simple photoperiod. It is suggested that the author take full consideration in designing the experiment in the future. The following contents are recommended for major modification.

Answers: Thank you for pointing out this. Although DLI is important for plant growth and development, previous studies have demonstrated that plant growth is highly significantly affected by photoperiod at the same total daily light input. This study mainly focused on effects of photoperiod interacted with nutrient solution concentration in lettuce.

Abstract:

Line 23-25: So, in the practical production, you choose biomass or nutritional quality? That doesn't seem to be the final conclusion of this study.

Answers: Thank you for pointing out this. The final conclusion of this study was described in abstract. We marked it with red color. Please find it in the revised manuscript.

Introduction:

The effects of photoperiod and nutrient concentration on plants and their importance should be introduced respectively, and more importantly is that why study the interaction between these two factors should be described in detail. Some introductions should be added for the amino acid, polyphenol, anthocyanin et al that several compounds analyzed in this expt.

Answers: Thank you for pointing out this. Some studies related to several compounds analyzed in this expt were added in introduction. We marked it with red color. Please find it in the revised manuscript.

Materials and methods

2.1 should be plant material and cultivation environment, and this section is not adequately described.

Answers: Thank you for pointing out this. The description of plant material had been added. We marked it with red color. Please find it in the revised manuscript.

Line 62: Growth chamber or plant factory? Described “plant factory” in abstract.

Answers: Thank you for pointing out this. This study based on the light demand of plant factory was performed.

Line 63: “the nutrient solution refreshed every 10 days”: The EC changes every day as plants grow. For the experiment which studied the concentration of nutrient solution, should the same EC and pH be maintained every day?

Answers: Thank you for pointing out this. The nutrient solution was refreshed every 10 days because the slight change of EC was detected for lettuce. pH was adjusted every two days.

Line 64: Even though it has table s1, but better indicate the ratio of NPK here.

Answers: Thank you for pointing out this. The ratio of NPK was indicated in Table S1.

Line 66-67: NSC is nutrient solution concentrations, should remove the concentrations in “combined with three concentrations of NSC”. Remove NSC in “(1/4, 1/2 and 3/4 NSC)”.

Answers: Thank you for pointing out this. The concentrations and NSC in line 66-67 were removed. We marked it with red color. Please find it in the revised manuscript.

Table 1: The design of this table is meaningless. The following table and figures also do not continue this representation. I recommend use the number of treatments: T1,T2…… or shorten the treatment name: 12-0.25x, 15-0.5x……..

Answers: Thank you for pointing out this. The treatment name was revised. Please find it in the revised manuscript.

2.5: No information on replication was found for this expt.

Answers: Thank you for pointing out this. The information of replication had already been indicated in line 135 of 2.5 section.

Results: 3.1: There is no description of this section in the M&M.

Answers: Thank you for pointing out this. The descriptions of 3.1 section were added in the M&M. We marked it with red color. Please find it in the revised manuscript.

Table 2&3: Are there no significant differences in other basic parameters of growth and development? For example: shoot & root length, leaf area, root biomass………..

Answers: Thank you for pointing out this. Plant dry weight is the most important growth parameter, and the other growth parameters is able to significantly assisted illustrate lettuce growth. However, it is regrettable that leaf length, leaf width and leaf area were not measured.

It doesn't make sense to leave two decimal places on the number of leaves. The decimal point & letters mark should be aligned & recommend to remove the standard error to show the data more clearly.

Answers: Thank you for pointing out this. The decimal places in Table 2&3 were revised. Please find it in the revised manuscript. The remined standard error might be more conducive to the data clearly.

For “ANOVA (F value)” recommend use the common notation: NS,*,**,***.

Answers: Thank you for pointing out this. The common notation of “ANOVA (F value)” was revised. Please find it in the revised manuscript.

Figure 1,2&3: The vertical coordinate already clearly represents the content of the data, so A, B, C… can be removed. Use standard representation for all units: mg/g → mg·g-1

Answers: Thank you for pointing out this. A, B, C… had been removed, and all units in Figures had been revised. Please find it in the revised manuscript.

Discussion & Conclusions

The interaction between photoperiod and nutrient concentration should be further discussed.

Answers: Thank you for pointing out this. The studies which were related to the interaction of photoperiod and nutrient concentration were few reports. The interaction of photoperiod and nutrient concentration how affects plant growth and development is still needed further research.

The importance of biomass and nutrient content can be discussed according to market demand, and then recommend one of the best results to the reader or grower.

Answers: Thank you for pointing out this. The result was recommended to the reader or grower in conclusion. We marked it with red color. Please find it in the revised manuscript.

Round 2

Reviewer 2 Report

Author has revised the manuscript as we suggested.